# Ferroptosis: Frenemy of Radiotherapy

**DOI:** 10.3390/ijms25073641

**Published:** 2024-03-25

**Authors:** Lisa Kerkhove, Febe Geirnaert, Inès Dufait, Mark De Ridder

**Affiliations:** Department of Radiotherapy, UZ Brussel, Vrije Universiteit Brussel, Laarbeeklaan 101, 1090 Brussels, Belgium; lisa.kerkhove@vub.be (L.K.); febe.geirnaert@vub.be (F.G.); ines.dufait@vub.be (I.D.)

**Keywords:** ferroptosis, radiotherapy, drug repurposing, normal tissue toxicity

## Abstract

Recently, it was established that ferroptosis, a type of iron-dependent regulated cell death, plays a prominent role in radiotherapy-triggered cell death. Accordingly, ferroptosis inducers attracted a lot of interest as potential radio-synergizing drugs, ultimately enhancing radioresponses and patient outcomes. Nevertheless, the tumor microenvironment seems to have a major impact on ferroptosis induction. The influence of hypoxic conditions is an area of interest, as it remains the principal hurdle in the field of radiotherapy. In this review, we focus on the implications of hypoxic conditions on ferroptosis, contemplating the plausibility of using ferroptosis inducers as clinical radiosensitizers. Furthermore, we dive into the prospects of drug repurposing in the domain of ferroptosis inducers and radiosensitizers. Lastly, the potential adverse effects of ferroptosis inducers on normal tissue were discussed in detail. This review will provide an important framework for subsequent ferroptosis research, ascertaining the feasibility of ferroptosis inducers as clinical radiosensitizers.

## 1. Introduction

In 2020, 19.3 million cancer patients were diagnosed worldwide, and 10 million cancer patients died. Globally, the impact of cancer is anticipated to escalate significantly, with an estimated 47% increase in cancer burden by the year 2040. Hence, cancer is still globally recognized as one of the most important health burdens. Therefore, continuous efforts and commitment towards the development of new cancer treatments are of utmost importance [1].

Radiation therapy (RT) is a cornerstone in cancer treatment. Approximately half of the cancer patients will receive palliative or curative RT. External beam RT, a commonly employed modality, exerts lethal effects on tumor cells by damaging critical components (most importantly, the DNA) within the cells. However, RT does not only target cancer cells but also healthy tissue, causing normal tissue complications that need to be avoided. Major technical improvements have been achieved over the past decade, including particle therapy, stereotactic body radiation therapy, intensity-modulated therapy, MRI-guided RT, etc. In complementation of maintaining the balance between tumor cell death and normal tissue sparing, the implementation of personalized treatment will be increasingly attainable [2,3].

RT has been described as inducing a multitude of cell fate decisions. In the past, the effects of RT were mainly attributed to induction of mitotic catastrophe, senescence, apoptosis, and necrosis. Other types of regulated cell death (RCD) have emerged as contributors to the efficacy of RT, namely necroptosis, pyroptosis, ferroptosis, autophagy, parthanatos, cuproptosis, and immunogenic cell death. Despite extensive research conducted on cell fate decisions following RT, the principal questions remain: which forms of RCD are predominantly induced following RT, and is it possible to intervene to enhance the efficacy of RT without causing toxicity. Numerous factors are important to take into consideration in determining the influence of RT on different cell fates: (i) cancer/cell type, (ii) phase of the cell cycle, (iii) mutational status, (iv) dose regimen and (v) microenvironmental features [4,5,6]. In comparison to tumor tissues, normal tissues tend to undergo senescence after RT [4]. Furthermore, apoptosis is mainly induced after low-dose RT, while high-dose RT will more likely provoke necrosis, although this classical view is disputed [5].

This review will focus on the role of one specific type of RCD in RT, namely, ferroptosis. Mounting evidence suggests that targeting ferroptosis has great potential in battling cancer. Due to its specific characteristics, it is expected that ferroptosis plays a major role in RT response. Consequently, combination strategies involving both modalities are intensively researched, resulting in promising preliminary data. On the other hand, studies have highlighted the involvement of ferroptosis induction in various fibrotic diseases. Since radiation-induced fibrosis is a well-known late side-effect of RT, clinical research into novel therapies combining ferroptosis inducers and RT must proceed cautiously, and toxicity should be considered. Here, we summarize the available data on the crosstalk between ferroptosis and RT and provide potential guidelines for translation to the clinic.

## 2. Ferroptosis

The first publications on ferroptosis date back to the early 2000’s. Ferroptosis differs from other types of RCD by its distinct morphology observed using electron microscopy. Additionally, the addition of classic apoptosis and necroptosis inhibitors, such as Z-VAD-FMK and necrostatin-1, were found ineffective in reversing the observed effects [7,8]. Microscopically, cells undergoing ferroptosis exhibit mitochondrial changes; shrunken mitochondria with an increased mitochondrial density, and decreased mitochondrial cristae [9]. Erastin was the first compound described to induce this new type of RCD in cells carrying an RAS mutation [10]. Soon after, additional compounds inducing ferroptosis were discovered, like RAS-selective ligand 3 (RSL3) [11]. However, there is still considerable controversy regarding the relationship between cell mutational status and ferroptosis sensitivity [12]. An important hallmark in ferroptosis research was the discovery of the inhibiting ability of iron-chelating agents, revealing the pivotal role of iron metabolism [11]. Nevertheless, the concept of ferroptosis was not established until 2012 [7]. Ever since, ferroptosis research has been thriving, particularly concerning the role of ferroptosis in cancer development, drug resistance, and therapy [13,14]. Nowadays, multiple ferroptosis-related gene signatures are being developed as a prognostic biomarker. Different research groups have met this goal for diverse tumor types, such as colorectal cancer [15], melanoma [16], and breast cancer [17]. Hence, understanding the role of different genes involved in the pathways of ferroptosis can facilitate the discovery of new therapeutic targets/biomarkers for the clinic.

Alongside iron metabolism, other cellular pathways are considered as main operators in ferroptosis. A brief summary will be given in the next paragraph; for more in-depth descriptions of the molecular mechanisms behind ferroptosis, we refer to other excellent reviews [18,19,20].

### 2.1. Terms and Conditions for Ferroptosis Execution

The hallmark of ferroptosis execution is the build-up of lipid peroxides exceeding a certain threshold. This threshold is reached when an imbalance between lipid peroxide formation and clearance by cellular defense systems is present. During this irreversible phase, the defense systems are unable to convert lipid peroxides into lipid alcohols, leading to the toxic accumulation of hydroperoxides and, eventually, membrane rupture [7,21].

Two different pathways are involved in the initiation of lipid peroxidation: an enzymatic lipoxygenase (LOX)/cytochrome P450 oxidoreductase (POR) dependent pathway and a non-enzymatic, Fenton-reaction initiating pathway [22,23,24,25]. Polyunsaturated fatty acid-containing phospholipids (PUFA-PL) are particularly prone to peroxidation due to their bis-allylic moieties [24]. After the generation of PL-radicals, interaction with oxygen and labile iron (Fe^2+)^ can take place. Leading to the generation of additional alkoxyl-/peroxyl-radicals and the accumulation of hydroperoxides. The alkoxyl-/peroxyl radicals again contribute to producing additional hydroperoxides, propagating the effect of the first peroxidation reaction. Hence, the driver of ferroptosis is a self-sustaining process referred to as autoxidation [26].

Modification of ferroptosis sensitivity in cancer cells is possible by intervening with the above-mentioned different players. Modifying lipid, iron, and mitochondrial metabolism, along with interrupting ferroptosis defense pathways, are the most extensively researched options.

### 2.2. Involvement of Lipid Metabolism in Ferroptosis

Considering that PUFA-PL are the most vulnerable to peroxidation because of their specific chemical structure, lipid metabolism plays a major role in ferroptosis sensitivity. The most prominent PUFAs present within the cells are arachidonic acids (AA) and adrenic acids (AdAs). Acyl coenzyme A (CoA) derivatives of these PUFAs are produced by acyl-coenzyme A synthetase long-chain family member 4 (ACSL4). After the production of these PUFA-CoA, lysophosphatidylcholine acyltransferase 3 (LPCAT3) esterifies them into phospholipids [27,28,29]. Interfering with the metabolism of mono-unsaturated fatty acids (MUFAs) appears to be a crucial factor in determining ferroptosis vulnerability. Non-oxidizable MUFAs counteract ferroptosis by replacing PUFAs in phospholipid bilayers. The generation of MUFA-CoA in this process is crucial and requires ACSL3 activity. Consequently, investigation into lipid metabolism exhibited that ACSL activity is a major regulator of ferroptosis (Figure 1A). Upregulated ACSL4 expression levels are linked to increased ferroptosis sensitivity, while inhibition of ACSL4 is connected to ferroptosis repression. Additionally, increased levels of ACSL3 expression are associated with ferroptosis resistance [27,28,29,30].

### 2.3. The Role of Iron Metabolism during Ferroptosis

Iron is the catalyst for the initiation of ferroptosis. It is also important in multiple biological processes, such as oxygen transport, ATP generation, and DNA biosynthesis [31]. Since an iron imbalance can interfere with a cell’s function, multiple mechanisms are present to maintain the iron balance. Two different forms of iron exist: the ferrous form (Fe^2+^), creating the active labile iron pool, and the ferric form (Fe^3+^), inactively bound to ferritin. Regulators of iron metabolism, which are involved in iron uptake, storage, utilization, and export, can influence a cell’s sensitivity to ferroptosis. The main operators in iron metabolism are transferrin receptors, ferritin, ferroportin, and nuclear receptor coactivator 4 (NCOA4) (Figure 1A) [31,32,33,34]. The uptake of iron is mainly regulated by Transferrin Receptor 1, which was recently dictated to be a biomarker for ferroptosis induction [35]. Furthermore, lactoferrin-bound iron (holo-lactoferrin) synergizes with ferroptosis-inducing therapeutics (RT) by increasing the total amount of iron uptake within cells [36]. Ferritin establishes the storage of iron in its inert form (Fe^3+^) within the cell. Research conducted in both cancer and healthy cells stated that suppression of the Ferritin heavy-chain enhances sensitivity to ferroptosis [37,38]. Additionally, ferritin export was reported to counteract ferroptosis [39]. Lastly, the degradation of ferritin (ferritinophagy) within the cell, mainly regulated by NCOA4, can increase the levels of labile iron, boosting ferroptosis [40].

### 2.4. Mitochondrial Involvement in Ferroptosis

Mitochondria are the main production sites of reactive oxygen species (ROS) within a cell. During the tricarboxylic acid (TCA) cycle, NADH is generated. This drives the proton gradient of the electron transport chain (ETC) to eventually generate energy (ATP). Formation of the superoxide radical arises at complex I and complex III of the ETC. Superoxide can sequentially be converted to hydrogen peroxide by superoxide dismutase. Hydrogen peroxide is the main driver of the Fenton reaction. Therefore, increased ROS production in the mitochondria is linked with more lipid peroxidation (Figure 1A) [41,42]. Additionally, ATP deprivation has been linked to AMP-activated protein kinase (AMPK) activation. This has been described as a negative regulator of ferroptosis since AMPK phosphorylates acetyl-coA carboxylase (ACC) and thereby inhibits PUFA synthesis, the prime target of lipid peroxidation [43,44,45]. However, despite mitochondria having ferroptosis-promoting properties, complex I and II seem to be key players in the reduction in ubiquinone (COQ) to ubiquinol (CoQH2). COQH2 is a major radical trapping antioxidant (RTA) involved in the defense against ferroptosis (see paragraph Ferroptosis defense systems). Moreover, mitochondria-depleted cells still retain the ability to undergo ferroptosis. Accordingly, the role of mitochondria in the ferroptosis process is still controversial and should be further elucidated [45].

### 2.5. Ferroptosis Defense Systems

#### 2.5.1. SLC7A11-GSH-GPX4 Axis

The best-known ferroptosis defense axis is mediated by glutathione peroxidase 4 (GPX4). GPX4 is considered to be the only GPX capable of catalyzing the reduction in toxic lipid peroxides to non-toxic lipid alcohols, thereby proving its importance as a ferroptosis defense system [46,47]. The active site of GPX4 contains a selenocysteine residue with radical trapping properties required for the functionality of GPX4 [48]. Glutathione (GSH) is the co-factor of GPX4. Consequently, the availability of GSH determines the achievable activity of GPX4. GSH is one of the most abundant antioxidants, composed of three amino acids: glycine, glutamate, and cysteine. The latter being the rate-limiting precursor. Hence, GPX4 activity is dependent on cysteine availability within cells (Figure 1A). Cysteine can be obtained via three different pathways. De novo biosynthesis via the transsulfuration pathway, however, accounts only for a limited amount of cysteine. Cysteine can also be released out of proteins during degradation. In addition to these two intracellular catalyzation pathways, most of the cysteine is taken up out of the extracellular environment via system xC- (SLC7A11 and SLC3A2), an antiporter responsible for the intracellular uptake of cystine coupled to the extracellular release of glutamate. Once taken up, cystine can be converted into cysteine and function as a precursor of GSH. Interfering with one of the components of the SLC7A11-GSH-GPX4 axis evidently makes cells more susceptible to ferroptosis [49,50]. Additionally, the first discovered small molecules able to induce ferroptosis (erastin and RSL3) are inhibitors of this principal pathway [7,10].

#### 2.5.2. BH4 Axis

Different GPX4-independent axes have been unraveled. Tetrahydrobiopterin (BH4) is an RTA involved in ferroptosis sheltering. BH4 biosynthesis is stimulated using GTP cyclohydrolase-1 (GCH1), which counteracts ferroptosis. Additionally, dihydrofolate reductase (DHFR) seems to play a crucial role in the regeneration of BH4, thereby stimulating its ferroptosis-inhibitory properties. Moreover, BH4 has the property to stimulate CoQ expression, which can be reduced to CoQH2, another RTA [51,52].

#### 2.5.3. The Ubiquinol (CoQH2) Axis

Ferroptosis suppressor protein 1 (FSP1), previously known as apoptosis-inducing factor mitochondrial 2 (AIFM2), has been identified as an important ferroptosis defense mechanism at a different subcellular location. Myristoylation of FSP1 enables transport to the plasma membrane, where FSP1 reduces CoQ to CoQH2 in an NADH-dependent manner, promoting RTA properties [53,54]. CoQH2 is not only active in the cytosol via BH4 and the plasma membrane through FSP1 but also in the mitochondria. Dihydroorotate dehydrogenase (DHODH) can produce RTA-active CoQH2 within the mitochondria to reduce the amount of mitochondrial lipid peroxidation [55].

An extensive investigation of interfering with all the different operators involved in ferroptosis aims to reduce the pathological effect of ferroptosis or exploit it as an anti-cancer mechanism [19,32].
Figure 1Cellular pathways influencing ferroptosis and impact of hypoxia on ferroptosis. (**A**) Initiation of ferroptosis is influenced by iron metabolism, mitochondrial metabolism, lipid metabolism, and different ferroptosis defense systems (including the SLC7A11/GSH/GPX4 axis). The levels of intracellular iron are regulated by iron import via transferrin receptors (TrfR), by iron storage in ferritin, and by iron export via ferroportin. In addition, stored iron can be released out of ferritin, increasing the labile iron pool (Fe^2+^) via a process called ferritinophagy, mediated by nuclear receptor coactivator 4 (NCOA4). The labile iron pool sequentially can interact with reactive oxygen species (ROS) generated in the mitochondria. This process is called the Fenton reaction and leads to the generation of hyperactive radicals. In turn, these radicals preferably interact with poly-unsaturated fatty acids (PUFAs), giving rise to lipid hydroperoxides (L-OOH-). The most prominent PUFA present within cells are arachidonic acids (AA) and adrenic acids (AdAs). Generation of PUFA-CoA is enzymatically initiated by linking Co-A to AA or AdAs by acyl-CoA synthetase long-chain family member 4 (ACSL4), followed by an esterifying step, catalyzed by lysophosphatidylcholine acetyltransferase 3 (LPCAT3). Synthesis of PUFA-CoA can be counteracted by acyl-CoA synthetase long-chain family member 3 (ACSL3), generating mono-unsaturated fatty acids (MUFA’s). Lipid hydroperoxides (L-OOH-) can be detoxified using GPX4. In order to substantiate this conversion, the most important co-factor glutathione (GSH) of GPX4 needs to be intracellularly present. The rate-limiting building block of GSH is cystine. Typically, it is taken up via system xC- (existing out of two subunits: SLC7A11 and SLC3A2). (**B**) Controversy exists about ferroptosis under hypoxic conditions. It is well-established that ROS levels are elevated under hypoxic conditions. Additionally, SLC7A11 has been described to be upregulated under low oxygen circumstances. Both are key players in ferroptosis; therefore, these increases interfere with the ferroptosis process under hypoxic conditions. HIF1α negatively regulates ferroptosis via inhibition of the iron metabolism, lipid metabolism, and SLC7A11-axis. HIF2α is positively correlated to ferroptosis by promoting PUFA synthesis and increasing the amount of iron within cells.
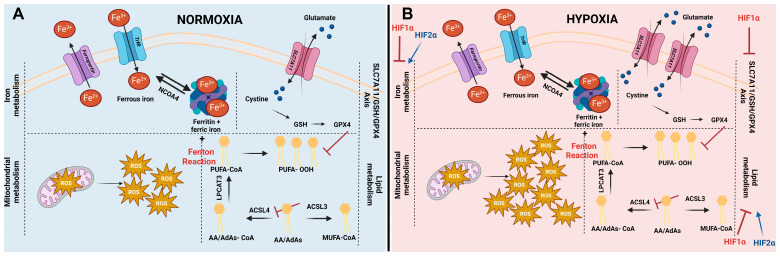


## 3. Ferroptosis and RT

### 3.1. Interplay between Ferroptosis and RT

Lately, the role of ferroptosis in RT efficacy has been under intensive investigation. This association has been observed in multiple cancer cell lines, including lung cancer, oesophageal cancer, breast cancer, fibrosarcoma, colorectal cancer, and melanoma [56,57,58,59]. Increased quantity of lipid peroxidation and enhanced prostaglandin-endoperoxide synthase 2 (PTGS2—a ferroptosis marker gene) or malondialdehyde (MDA) levels were observed in different cell lines after single dose ionizing radiation (IR). Additionally, typical morphological features related to ferroptosis were observed in lung cancer cells after exposure to radiation [58]. Diverse underlying courses of action were reported. Lei et al. hypothesized that increased levels of ROS generated during IR increase the plausibility of interaction with PUFA-PL. Additionally, elevated ACSL4 levels were described after IR, marking the interference with lipid metabolism as one of the key regulators of ferroptosis induction after RT. However, compensation mechanisms within these cells were observed by upregulation of GPX4 and SLC7A11 [58], which are involved in the defense against ferroptosis, as described above. In contrast, Lang et al. determined that increased levels of lipid peroxidation were present due to elevated expression levels of the Ataxia-Telangiectasia mutated gene (ATM) after IR, which has the capacity to inhibit SLC7A11 (Figure 2). Furthermore, the inhibition of SLC7A11 is considered to also be responsible for the crosstalk between RT and immunotherapy [56]. Ye et al. noticed that ferroptosis inducers were able to synergize with cytoplasmic IR; however, no contemplation was observed after nuclear irradiation [57].

Despite ongoing research, a lot of questions remain unanswered. It is important to determine the role of other pathways involved in RT-induced ferroptosis, as differential responses are observed in a variety of cancer cells upon exposure to ferroptosis inducers. Generally, it is accepted that p53 is upregulated after IR. P53 upregulation has been described as a dual regulator of ferroptosis. The positive regulation is related to the suppression of SLC7A11, while the negative impact of p53 was linked to the translocation of dipeptidyl-peptidase-4 (DPP4) to the nucleus, where it loses its ability to induce lipid ROS [60,61]. Collectively, this undeniably demonstrates that the exact role of p53 is context-dependent. 

### 3.2. Effect of Hypoxia on Ferroptosis

Tumor hypoxia is a common feature of solid tumors and is recognized as one of the principal causes of clinical radioresistance. As a result, modification by hyperbaric oxygen and nitroimidazoles has been exploited to overcome hypoxic resistance. Unfortunately, every effort remained largely ineffective for clinical application [62,63]. The association between hypoxia and ferroptosis is under debate. It is well-known that hypoxia results in an increase in intracellular ROS due to the prolongation of the lifetime of the ubisemiquinone radical, a molecule generated at complex III of the mitochondria. Additionally, inducible nitric oxide synthase (iNOS), a gene responding to hypoxia, leads to increased levels of nitric oxide (NO) [41]. These elevated ROS levels theoretically suggest an increase in ferroptosis. However, different hypoxia-related genes have different effects on the sensitivity of cells to ferroptosis [64].

The most prominent genes expressed under hypoxic conditions are of the hypoxia-inducible factor (HIF) family. HIFs are accountable for the adaptations of cells lacking oxygen via transcriptional regulation of genes possessing a hypoxia response element (HRE). Active HIF consists of two subunits, HIF1α, HIF2α, or HIF3α bound to HIF1β. HIF1β is constitutively expressed, while the activity of HIFα is dependent on the supplied oxygen. Under normoxic conditions, oxygen attaches to HIFα, which is subsequently hydroxylated by HIF prolyl hydroxylases (PDH1, PDH2, and PDH3). Hydroxylated HIFα conducts a complex formation with von Hippel Lindau, facilitating ubiquitination and proteasomal degradation of HIFα. However, under hypoxic conditions, these complex interactions do not take place, and HIFα can dimerize with HIF1β and become translationally active in the nucleus [65].

Hypoxia negatively regulates ferroptosis in oral squamous cell carcinoma, cervical cancer cells, hepatocellular carcinoma, macrophages, and osteoclasts [66,67,68,69,70]. Nonetheless, the exact mechanism of action is cell type dependent. For instance, HIF1α activation impeded the process of ferritinophagy in macrophages via downregulating NCOA4. This effect was not observed in osteosarcoma cells [69]. Additionally, under hypoxic conditions, HIF1α inhibited ferritinophagy in osteoclasts, leading to immense bone resorption due to their overactivation [70]. Hence, HIF1α has a negative impact on ferritinophagy; however, this effect seems to be cell-type dependent (Figure 1B).

In oral squamous cell carcinoma, it was discovered that the HIF1α expression level has a bidirectional link to the period 1 gene (PER1) expression. PER1 can attach to HIF1α, initiating the degradation of the protein. Additionally, HIF1α can bind to the promoter region of PER1, leading to inhibition of the transcription of this gene. Additionally, PER1 expression negatively correlates to GPX4 expression, suggesting that under hypoxic conditions, reduced PER1 levels in tumor tissues activate GPX4, leading to resistance against ferroptosis [68] (Figure 1B).

On the contrary, epigenetic regulation of genes is also involved in ferroptosis resistance. In cervical cancer cells, HIF1α is epigenetically regulated by lysine (K)-specific demethylase 4A (KDM4A), which, in turn, was found to be upregulated under hypoxic conditions. KDM4A counteracts the gene-silencing activity of methylation, leading to enlarging levels of HIF1α and, thereby, its ferroptosis inhibitory properties [67]. Lastly, in hepatocellular carcinoma, a downregulation of another epigenetic regulator, methyltransferase 14 (METTL14), occurs under hypoxic conditions. METTL14 was discovered to play a key role in decreasing the expression levels of SLC7A11. Therefore, hypoxic conditions in hepatocellular carcinoma restrict ferroptosis (Figure 1B) [66]. Further, Wen et al. constructed a prognostic signature containing ferroptosis-related genes and hypoxia-related genes for hepatocellular carcinoma patients. The patients with a high prognostic signature exhibited a ferroptosis repressive status and a high hypoxic environment. Overall, these patients had shorter overall survival, higher mortality, advanced TNM staging, and poor tumor differentiation [71]. Moreover, it was uncovered that HIF1α led to ferroptosis resistance by altering the lipid metabolism. Under HIF1α activation, more lipid droplet formation is present, declining the levels of PUFAs and the prospect of lipid peroxidation (Figure 1B) [72].

As opposed to the influence of HIF1α on lipid metabolism, Zou et al. provided evidence that HIF2α activated hypoxia-inducible lipid droplet-associated protein (HILPDA), enriching the amount of PUFA [73] (Figure 1B). HIF2α has also been implicated in the regulation of iron metabolism. Mainly by increasing the intracellular iron absorption and uptake, thereby rendering colorectal cancer cells and enterocytes sensitive towards ferroptosis (Figure 1B) [74,75].

In summary, while there is ongoing controversy around ferroptosis under hypoxic conditions, it is evident that sensitivity towards ferroptosis under hypoxic conditions varies depending on the specific situation and cell type. Overall, HIF1α has a negative impact on ferroptosis induction, while HIF2α positively regulates ferroptosis [64]. Hence, further research into this area is desired before ferroptosis inducers can be fully exploited in solid tumors.
Figure 2The interplay of radiation and drug repurposing candidates with the fundamental molecular mechanisms in ferroptosis induction. This figure depicts the most important molecular mechanisms implicated in ferroptosis, including lipid metabolism, iron metabolism, mitochondrial metabolism, and the different defense systems. Additionally, the interference of radiation exposure on these networks is symbolized by a lightning symbol. Irradiation encompasses upregulated levels of ROS, augmented occurrences of DNA double-strand breaks, amplified mitochondrial ROS, upregulation of ACSL4, and activation of ATM and p53. Upregulated levels of ROS are postulated to be responsible for more lipid peroxidation via the Fenton reaction. Concurrently, elevated levels of ACSL4 are accountable for the increased generation of PUFAs, particularly susceptible to lipid peroxidation. Furthermore, elevated levels of p53 and ATM increase susceptibility to ferroptosis, as they negatively regulate SCL7A11, a pivotal molecule in the ferroptotic defense. Additionally, ferroptosis-inducing mechanisms attributed to the ten described drug repurposing candidates (Section 4.2) are delineated. Dashed arrows denote an activation/increase/stimulation; blunt arrows denote an inhibition within the depicted pathways.
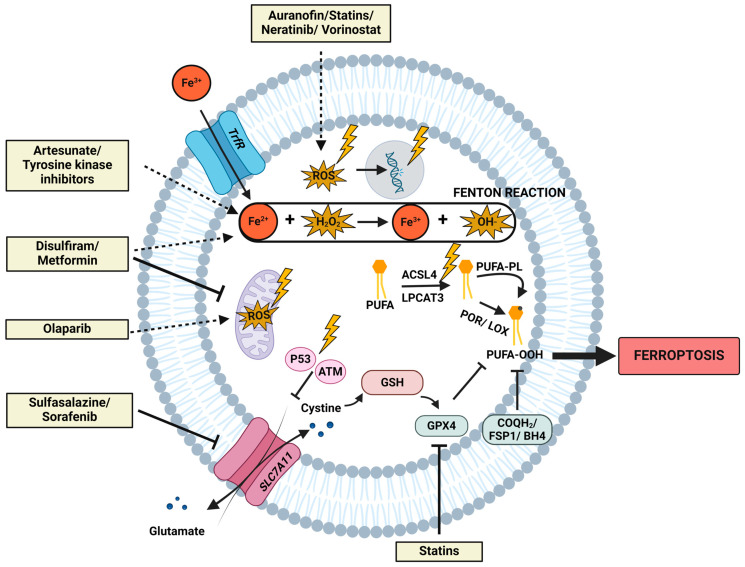


## 4. Ferroptosis Inducers and the Role in RT Effectiveness

The interplay between ferroptosis inducers (FINs) and RT has been established. A multitude of FINs, including erastin, sulfasalazine, sorafenib, RSL3, ML162, IKE, and FIN56, can synergize with RT in vitro and in vivo [56,57,58,59]. Moreover, the first clinical link between ferroptosis and radiotherapy was elucidated by Lei et al. The degree of ferroptosis induction in samples of oesophageal cancer patients showed a correlation with disease-free survival in response to RT [58].

Well-established radiosensitizers, like poly ADP-ribose polymerase (PARP)-inhibitors, are nowadays considered to enhance radio-efficacy via ferroptosis stimulation, in addition to their traditional mechanism of action, which involves inhibiting DNA repair within cancer cells. PARP-inhibitor Niraparib activates cyclic GMP-AMP synthase (cGAS), leading to increased activating transcription factor 3 (ATF3) expression and, consequently, the inhibition of SLC7A11, thereby directly stimulating ferroptosis [76]. Moreover, newly discovered radiosensitizers, like nanoparticles, are being associated with properties that initiate ferroptosis. AGuIX possesses these properties by inhibiting nuclear factor erythroid-2 related factor 2 (NRF2), resulting in decreased SLC7A11 expression and GSH [77]. Hence, investigation into hitherto studied radiosensitizers regarding their relation to ferroptosis may yield new insights.

### 4.1. Drug Repurposing for Radiosensitizing Properties via Ferroptosis

Drug repurposing (DR) has attracted a lot of interest over the past decades. It is referred to as studying the efficacy and safety of approved drugs for a certain disease in the treatment of a non-approved condition. Overall, the interest in DR was sparked because it is associated with lower overall costs, reduced risk of failure, and minimized developmental time [78].

A striking DR strategy was the use of thalidomide, initially discovered for the treatment of morning sickness and as a sedative. Nowadays, thalidomide is repurposed as a drug with anti-cancer properties and indispensable in the treatment of multiple myeloma patients [78,79]. Moreover, the use of FDA-approved drugs as FINs is under intensive investigation. Two pharmacological agents, sulfasalazine and sorafenib, were initially developed for the treatment of rheumatoid arthritis and hepatocellular carcinoma, respectively. However, it has been elucidated that these drugs contain system xC-inhibitory capacities, inducing ferroptosis within a range of cancer cells [80]. The link with RT will be described in-depth in the following paragraphs. Despite advances in DR, the quantity of candidate molecules remains limited. In the following section, we provide an overview of what we believe to be the ten most promising drugs to which radiomodulating features have been ascribed and ferroptotic properties have been examined (Table 1).

### 4.2. Drugs Containing Radiosensitizing Properties Linked to Ferroptosis Induction

1.Sulfasalazine

Sulfasalazine (SSZ), an FDA-approved drug for the treatment of rheumatoid arthritis and inflammatory bowel diseases (IBD), is recognized as an xCT (encoded by SLC7A11) inhibitor and therefore acknowledged as a FIN (Figure 2).

The radiomodulatory properties of SSZ have been established in various cancer cells under normoxic conditions, including melanoma cells, glioma cells, osteosarcoma cells, ovarian cancer cells, and non-small cell lung cancer cells (NSCLC) [56,58,81,82]. Additionally, SSZ had radiosensitizing effects on hypoxic colorectal cancer cells [59]. In melanoma and glioma cells, the mechanisms responsible for the observed effects were attributed to increased DNA damage following enhanced ROS levels and cell cycle arrest [81,82]. Nevertheless, the link with ferroptosis induction was not studied. More recent papers have established ferroptosis induction as an underlying mechanism for radiosensitization by SSZ [56,58,59]. Exploration of SSZ as a radiosensitizer in clinical trials is limited. The combination of SSZ treatment and RT is being investigated in one trial for glioblastoma patients [83], and results are eagerly awaited.

2.Disulfiram

Disulfiram (DSF), an inhibitor of aldehyde dehydrogenase, is commonly used in the clinic as Antabuse [84]. Recently, disulfiram treatment has been linked to novel types of programmed cell death, such as cuproptosis and ferroptosis (Figure 2).

Radiomodulatory properties of DSF have been explored in a number of cancer cells, including glioblastoma, pancreatic cancer, atypical children brain tumors, breast cancer cells, neuroblastoma cells, glioma cells, etc. [85,86,87,88]. Treatment with DSF radiosensitized some of the aforementioned cancer types [85,88]. Additionally, DSF exhibited radioprotective effects on normal tissues, making this molecule favorable in a radiobiological setting [87]. Mechanisms underlying the observed radiosensitizing effects have been mainly attributed to increased DNA damage, cell cycle arrest, apoptosis induction, autophagy induction, cell adhesion molecule signaling, induction of oxidative stress, and proteasome inhibition [85,86,88]. Of note, the evidence linking the radiomodulatory effect of DSF to ferroptosis induction is currently restricted to a single publication [89].

3.Sorafenib

This multi-kinase inhibitor (Nexavar) is FDA-approved for the treatment of unresectable hepatocellular carcinoma, advanced renal cell carcinoma, and locally recurrent or metastatic thyroid carcinoma. In the early years following its discovery, sorafenib’s anti-cancer properties were attributed to induction of apoptosis, inhibition of angiogenesis, and inhibition of proliferation. This targeted therapy inhibits the function of the RAF/MEK/ERK pathway and other tyrosine kinases, like VEGF and PDGF, involved in tumor proliferation and angiogenesis [90]. Irrespective of the initially discovered mechanisms, it was later found that sorafenib has the ability to induce ferroptosis within hepatocellular carcinoma cells [91]. Shortly after, sorafenib was described as a system xC- inhibitor, containing similar ferroptosis-inducing properties as erastin (Figure 2) [92].

Over a decade ago, the first studies demonstrating the radiosensitizing effects of sorafenib were published in hepatocellular carcinoma, colorectal cancer, and head and neck squamous cell carcinoma [93,94,95,96]. The underlying mechanisms mainly accounted for inhibition of the RAS/RAF/MEK/ERK pathway, decreased DNA double-strand break repair, G2/M cell cycle arrest, apoptosis induction, ROS generation, invasion and metastatic capacity inhibition, and obstruction of angiogenesis [93,94,95,96]. Hitherto, only 1 paper has linked the radiosensitizing effects of sorafenib to ferroptosis induction so far. Lei et al. demonstrated that in vitro treatment with sorafenib radiosensitized HT-1080 fibrosarcoma cells by decreasing the levels of GSH, and this effect was reversed by the addition of ferrostatin-1. In vivo confirmation of these results was performed in a xenograft model [57].

### 4.3. Drugs with Ferroptosis Inducing Capacities and Radiosensitizing Properties

4.Auranofin

Auranofin (AF), an FDA-approved drug auranofin for the treatment of rheumatoid arthritis, is under investigation as an anti-cancer medicine and FIN (Figure 2).

Radiosensitizing effects of AF have been reported. AF radiosensitized both normoxic and hypoxic breast cancer cells by inhibiting thioredoxin reductase (TrxR), a major antioxidant in cells, leading to enhanced levels of ROS and DNA damage in vitro. Furthermore, hypoxic cancer cells were sensitized to IR by a decrease in mitochondrial oxygen consumption. Despite promising in vitro results, no radiosensitizing effect of AF could be detected in vivo. Combining AF with other therapeutic agents, like buthionine sulfoximine (BSO), enabled radiosensitization of breast cancer in vivo [97]. Other studies have also employed a combination of AF with BSO and SSZ as radiomodulators. This combination strategy led to increased DNA damage, which was attributed to be the cause of the observed radiosensitization [98]. Moreover, AF has the potential to radiosensitize pancreatic cancer cells when combined with ascorbate. Ascorbate is capable of increasing the intracellular levels of hydrogen peroxide (H_2_O_2_). Combination with AF leads to a more pronounced elevation of H_2_O_2_ due to its inhibitory properties on TrxR activity [99]. Clinical investigation into DR of AF is limited to combining AF with chemotherapeutics.

5.Metformin

Repurposing metformin, a standard of care in the treatment of diabetes, as an anti-cancer drug has attracted attention in recent years. The radiosensitizing capacity of metformin was discovered a decade ago and can be linked to diverse processes. Metformin radiosensitized colorectal cancer cells, as well as breast and fibrosarcoma cells, by inhibiting complex I of the mitochondria, leading to a depletion in the levels of ATP and the activation of AMPK. Activated AMPK inhibits the PI3K/Akt/mTOR pathway, a known pro-tumorigenic pathway, by inhibiting its downstream targets (S6K1 and 4EBP1) [100,101]. The combination of metformin with RT was at least as efficient as the standard of care combination in colorectal cancer: 5-FU + RT [100]. Moreover, metformin was able to radiosensitize hypoxic colorectal cancer cells by inhibiting complex I of the mitochondria, leading to oxygen-sparing in vitro and in vivo [102]. Additionally, metformin seems to selectively kill cancer stem cells, which are the most radioresistant type of tumor cells [101]. Subsequently, the drug also alters the immune status [103]. Additionally, metformin has been linked to the induction of ferroptosis by modulation of several pathways, such as SLC7A11, GPX4, iron metabolism, etc., in breast cancer (Figure 2).

6.Artesunate

Artesunate is globally used as an anti-malarial drug. During the last decade, a lot of research has focussed on its anti-cancer properties. Essentially, artesunate is predicted to induce oxidative stress via its endoperoxide bridge, which is cleaved in the presence of ferrous iron [104]. More recently, artesunate treatment has been linked to altered iron homeostasis within cells. Therefore, the drug is acknowledged as a FIN (Figure 2).

Radiosensitizing features have been ascribed to artesunate over a decade ago and are linked to increased levels of ROS and DNA damage, influencing the cell cycle and apoptosis induction in glioma cells, cervical cancer cells, and oesophageal cancer cells [104,105,106]. The usual suspects influencing RT efficacy seem to be affected by this drug.

7.Statins

This group of drugs is one of the most commonly used drugs worldwide because of its cholesterol-lowering capacities. Statins inhibit 3-hydroxy-3-methyl-glutaryl-CoA (HMG-CoA) reductase, a crucial enzyme in the de novo synthesis of cholesterol. Therefore, these drugs are used to prevent hypercholesterolemia and cardiovascular diseases. Alongside its HMG-CoA reductase inhibitory feature, statins can target multiple other cellular substances and are consequently being repurposed as anti-cancer drugs [107]. Ferroptosis initiation upon statin treatment was shown in a range of cell types (Figure 2).

During the last decade, statins have been acknowledged as radiosensitizers. Clear radiosensitizing effects have been described in inflammatory and triple-negative breast cancer cells, prostate cancer, and melanoma. Various retrospective analyses proved that statin-use in combination with RT was linked to increased local recurrence-free survival in breast cancer patients, increased relapse-free survival in prostate cancer patients, and improvements in disease-free survival, local recurrence, cancer-specific mortality, and overall survival in rectal cancer patients [108,109,110]. Underlying mechanisms attributed to the radiosensitizing properties of these lipid-lowering drugs include inhibition of DNA repair, inhibition of proliferation, induction of senescence, and induction of apoptosis [111,112]. Recently, the hypoxia-alleviating properties of statins have also been uncovered, and the influence of decreased oxygen consumption after statin treatment in prostate cells has been linked to radiosensitivity. However, no positive correlation could be found [113].

8.Vorinostat

Vorinostat is a class I histone-deacetylase (HDAC) inhibitor that is FDA-approved for the treatment of cutaneous T-cell lymphoma. HDAC inhibitors increase the amount of histone acetylation, shifting the chromatin to a more relaxed state, subsequently promoting the expression of tumor suppressor genes and influencing non-histone proteins [114]. The first link between ferroptosis and vorinostat has recently been established in lung cancer cells during combination treatment with erastin (Figure 2) [115].

Since the development of HDAC inhibitors as an anti-cancer therapy, the radiomodulatory properties of vorinostat have been under investigation. Vorinostat treatment increased radiosensitivity in lung cancer cells, breast cancer cells, and colorectal cancer cells [116,117,118], which was attributed to increases in DNA damage linked to dysregulated DNA double-strand break repair and increased ROS generation. Additionally, cell cycle arrest and apoptosis induction were observed [116,117,118]. One study included not only normoxic conditions but also hypoxic conditions [116], which is considerably interesting from a radiobiological standpoint. Additionally, the influence of vorinostat was evaluated on normal tissue to assess its potential to enhance the therapeutic window. No radiosensitizing effect was observed in lung fibroblasts after vorinostat treatment despite induction of DNA damage and cell cycle arrest [118].

9.Olaparib

Olaparib is the most widely used PARP inhibitor and is FDA-approved for several indications, including breast cancer, ovarian cancer, prostate cancer, fallopian tube cancer, peritoneal cancer, and pancreatic cancer [119]. The involvement of PARPs in multiple cellular processes, like metabolic regulation, programmed cell death, transcriptional regulation, DNA methylation, and DNA repair, is well established [120]. PARP is one of the main effector molecules in the repair of DNA single-strand breaks. By inhibiting PARP, single-strand breaks can turn into double-strand breaks during replication. Therefore, the use of PARP inhibitors is extremely interesting in cancer with homologous recombination repair defects, one of the main mechanisms for repairing DNA double-strand breaks. However, recently, the effects of PARP-inhibitors were attributed beyond influencing DNA repair mechanisms and linked to the induction of ferroptosis (Figure 2) [119,120,121].

A decade ago, the first reports on olaparib as a radiosensitizer were published. Nowadays, a variety of reports can be found combining olaparib with X-irradiation, proton therapy, etc. Treatment with olaparib has radiosensitized a multitude of cancer cells, including neuroblastoma [122], oesophageal carcinoma cells [123], cervical cancer [124], and colorectal cancer cells [125]. To date, the molecular mechanisms attributed to the radiosensitizing effects include cell cycle arrest in the G2/M phase, increased DNA damage, persistent DNA damage, and the induction of senescence [122,123,124,125]. Currently, 29 clinical trials are investigating olaparib in combination with IR (ClinicalTrials.gov (accessed on 30 January 2024)). The results of the first clinical trials claim that this combination therapy is feasible, with limited adverse events [126].

10.Tyrosine kinase inhibitors (Lapatinib and Neratinib)

1 out of 40 genes in cells encode protein kinases, which are proteins involved in the development and progression of cancer [127]. Nowadays, over 50 kinase inhibitors with specific targets have been developed as targeted anti-cancer therapy. Of interest in this review are two tyrosine kinase inhibitors, Lapatinib and Neratinib, developed for the treatment of HER-2+ breast cancer [128]. 20% of breast cancers are HER-2+, leading to a constant activation of HER-2 and its downstream pathways, namely PI3K/AKT and RAS/RAF/MEK/MAPK, which are involved in proliferation and cell death escape mechanisms [128]. Ferroptosis induction following lapatinib in combination with siramesine or a single treatment with neratinib has been described in diverse cancer types (Figure 2) [129,130,131].

The radiomodulatory properties of Lapatinib were first described in 2010. HER2+ and EGFR+ breast cancer cells were radiosensitized using lapatinib in a fractionated radiation treatment regimen. However, the exact underlying mechanism was not unraveled [132]. Lapatinib was confirmed to be a radiosensitizer in two different breast cancer cell lines. Mechanistically, the radiomodulatory properties were attributed to prolonged DNA damage, induction of apoptosis, and senescence [133]. Neratinib is recognized as a radiosensitizer in head and neck squamous cell carcinoma. The increase in sensitivity towards RT was attributed to increased levels of apoptosis [134]. Currently, 19 clinical trials are investigating the combination of lapatinib with RT (ClinicalTrials.gov (accessed on 30 January 2024)).

Conclusively, a multitude of underlying mechanisms have been attributed to the radio-effects of the different drugs. Regardless of the comprehensive pre-clinical studies, the role of ferroptosis induction has only been investigated in three out of the ten selected drugs and remains to be clarified in the other seven.
ijms-25-03641-t001_Table 1Table 1FDA-approved drugs acknowledged to induce ferroptosis and their molecular mechanisms.DrugFDA Approved ConditionFIN in Cancer TypesMolecular Mechanism1. SulfasalazineRheumatoid arthritis and IBDBreast [135], Colon [136], Brain [137]Inhibitor of xCT, light subunit of system xC- → decreases levels of GSH, less active GPX4, more ROS and more lipid peroxidation2. DisulfiramAntabuseBrain [89], Liver [138], Naso-pharyngeal [139]Increases levels of ROS, impairs mitochondrial homeostasis, increases iron levels, increases lipid peroxidation3. SorafenibHepatocellular and renal cell carcinomaLiver [91], Kidney [140], Lung [140], Colon [140], Pancreas [140], Skin [140]System xC- inhibitor → decreases levels of GSH, leading to less active GPX4, more ROS and increases lipid peroxidation4. AuranofinRheumatoid arthritisCervix [141], Liver [142], Brain [99]Inhibits TrxR, increases ROS, increases lipid peroxidation, activates hepcidin5. MetforminDiabetesBreast [143,144,145]Inhibits autophagy, upregulates miR-324-3p and thereby inhibits GPX4, increases iron levels, and lipid ROS, decreases SLC7A11 stability via inhibition of UFMylation6. ArtesunateAnti-malaria agentBrain [146], Lymphatic system [147,148], Liver [149]Modulates the p38/ERK pathway, impairs STAT3 signalling, induces ER stress, activates the ATF4-CHOP-CHAC1 pathway, increases intracellular iron7. StatinsCardio-vascular diseasesBreast [150], Non-tumorigenic tissue [151]Inhibits HMG-CoA reductase → inhibits mevalonate pathway → inhibits biosynthesis of GPX4. Increases the levels of iron, increases the levels of ROS and lipid peroxidation, inhibits NRF2, SLC7A11, and GPX48. VorinostatCutaneous T-cell lymphoma Lung [115]Increases the amount of lipid peroxidation/ROS, inhibits xCT9. OlaparibBRCA mutated cancerOvarian [120,121]Activates P53, inhibits SLC7A11, decreases levels of GSH, increases levels of lipid peroxidation, suppression of SCD110. Tyrosine kinase inhibitorHER2+ Breast cancerBreast [129,130], Brain [131], Lung [131]Increases the levels of iron (Neratinib), increases ROS, increases lipid peroxidation increases transferrin expression, decreases ferroportin expression, degradation of heme-oxygenase (Lapatinib + Siramesine)

## 5. Adverse Events of Ferroptosis Inducers as Radiosensitizers

RT is a highly effective curative treatment; although, unfortunately, normal tissue toxicity is a major issue and the prime concern for radiation therapists. Hence, an investigation into processes underlying normal tissue toxicity is of supreme importance and remains understudied. This is evidenced by the fact that only one radioprotector, namely amifostatine, is FDA-approved for the treatment of head- and neck cancers [152]. Therefore, in-depth knowledge of the processes involved in normal tissue toxicity development and the expected impact of radiosensitizers could improve treatment options. This approach aims to focus on a window of acceptable radiation doses, limiting normal tissue toxicity and maximizing tumor cell killing [153,154].

Radiation-induced normal tissue toxicity can be divided into acute toxicity and late/chronic toxicity. Acute toxicity develops during radiation and within weeks after the treatment regimen, while late toxicity can occur 4 months to even years after completing an RT treatment course. These timing differences can be associated with the proliferative speed of the affected cells/organs. Rapid proliferating cells will quickly accumulate damage upon radiation, leading to acute effects. On the contrary, slowly to non-proliferating cells are responsible for late/chronic toxicity, which bears a higher burden on patients [153,154,155].

A major late toxic effect is the development of radiation-induced fibrosis (RIF), which originates in different tissues of the human body, such as skin, lungs, genitourinary tracts, subcutaneous tissues, and gastrointestinal tissues. RIF is characterized by an inflammatory reaction of the cells to the trauma induced by radiation, which is further amplified over time. Normal tissue within the radiation frame that gets injured initiates the production of pro-inflammatory cytokines. These cytokines attract immune cells to the site of injury, including monocytes capable of differentiating into macrophages. Macrophages have the capacity to secrete-platelet-derived growth factor (PDGF), which attracts fibroblasts out of the vascular or stromal compartment. Moreover, macrophages typically secrete transforming growth factor β (TGF-β), a key contributor of RIF. TGF-β is competent at differentiating fibroblasts into myofibroblasts and stimulates epithelial-to-mesenchymal transition. Sequentially, these myofibroblasts are accountable for the deposition of excessive extracellular matrix components, such as fibronectin, collagen, and proteoglycans. This excessive production of extracellular matrix can eventually lead to functional and cosmetic impairments of the affected tissues, decreasing the quality of life in patients [155].

Recently, a tight link between ferroptosis and fibrosis has been revealed. As described by Linkermann et al., a tight bidirectional connection between RCD and inflammation exists. It is well-established that specific types of RCD, including ferroptosis, can activate the immune system and initiate an inflammatory reaction [156]. Therefore, ferroptosis induction has been implicated in the pathogenesis of fibrogenesis in diverse organs, including the liver, kidney, myocardium, gastrointestinal system, salivary glands, pancreas, prostate, and lung [157,158,159]. The ferroptotic mechanisms underlying the development of fibrosis in these organs have been thoroughly revised elsewhere [157,159]. Albeit ferroptosis induction has been extensively linked to fibrosis, knowledge about the role of ferroptosis in RIF and acute tissue toxicity is still limited.

### 5.1. Interplay between Ferroptosis and Total Body Irradiation

It is extensively described that total body irradiation (TBI) can induce normal tissue toxicity in multiple tissues within the body. As mentioned above, only one radioprotector is FDA-approved. Despite approval, the use of amifostine in the clinic is limited due to severe toxicity. Zhang et al. developed a novel radioprotector (compound 5) with the same potency as amifostine in alleviating radiation injury but replaced the polyamine backbone with a polycysteine backbone, reducing the toxic side-effects. The observed radioprotecting effects were linked to the inhibition of ferroptosis via depletion of lipid ROS. Compound 5 increased the levels of GSH and GPX4 and decreased the activity of NOX1, an enzyme involved in ROS production [160]. Hence, the first association between radioprotective effects and ferroptosis was established.

Thermozier et al. described the importance of different cell death pathways for mitigating the effects of TBI on normal tissues. Apoptosis and necroptosis are two well-acknowledged pathways involved in normal tissue injury after TBI but seem to be time-dependent. Apoptosis was demonstrated to be initiated within 24 h after TBI, while necroptosis initiation can only be detected 48 h after TBI. The kinetics of ferroptosis induction have not been elucidated yet. However, ferroptosis seems to be optimally activated 24 h after TBI, indicated by increased expression levels of ACSL4 and decreased expression levels of GPX4 [161].

### 5.2. Link between Ferroptosis and Radiation-Induced Skin Injury

The first link between ferroptosis and radiation-induced skin injury (RSI) was established by Xue et al., who demonstrated the importance of BH4 availability (one of the key regulators counteracting ferroptosis) during radiation. However, the link between BH4 and ferroptosis was not investigated as this study focused on the function of BH4 in generating NO. Decreases in BH4 activity can lead to the uncoupling of NOS, resulting in the generation of reactive nitric oxide species (RNS) and ROS. Following irradiation (20 Gy), decreased levels of BH4 and GTP cyclohydrolase I (GCH1), the rate-limiting enzyme in de novo generation of BH4, were observed in both skin cells and patient samples. Overexpression of GCH1 or BH4 supplementation resulted in a reduced number of DNA double-strand breaks, depolarization of mitochondria, and endoplasmic reticulum dysregulation in cultured cells. Furthermore, this treatment in vivo reduced skin appendages and epidermis depth after irradiation (45 Gy) [162] (Figure 3). Hence, the first link between BH4 and skin injury was established. 

Additionally, a link between ferroptosis and ultraviolet (UV)-RSI was uncovered. Exposure of mouse skin to UV-radiation led to increased epidermal and dermal thickness and increased density of skin collagen. The aforementioned skin injury was linked to increased levels of lipid peroxidation (4-HNE and 8-OHdG) and increased iron. Despite these increases, the cells appeared not to undergo ferroptosis due to the increased expression and protection of GPX4. Nonetheless, this study described the capability of ferrostatin-1 and nicotinamide mononucleotide (NMN) to diminish the severity of RSI [163] (Figure 3).

### 5.3. Role of Ferroptosis in Radiation-Induced Lung Injury

Li et al. first described the role of ferroptosis in the development of radiation-induced lung fibrosis (RILF). They observed that ferroptosis is important during fibrosis development, as demonstrated by treating mice with liproxstatin-1, a ferroptosis inhibitor, during RILF development. 20 weeks after irradiation (15 Gy), a clear decrease in GPX4 expression was observed. Additionally, the fibrotic severity was scored higher, and the expression levels of hydroxyproline, a major component of collagen, were increased in the lungs of irradiated mice. ROS levels and pro-inflammatory cytokines, especially TGF-β, were elevated after radiation of the lungs. Intervening with liproxstatin-1 during irradiation counteracted decreases in GPX4 levels, inhibiting fibrotic remodeling. This effect could be attributed to activation of NRF2. As a consequence, lower levels of ROS were present, leading to a decrease in the secretion of TGF-β, alleviating the radiation-induced fibrotic remodeling [164] (Figure 3).

Additionally, Li et al. unraveled the role of ferroptosis during acute radiation-induced lung injury (RILI). Ferroptosis was again a crucial key player in the development of structural lung damage and hemorrhage 15, 30, and 60 days after radiation with 15 Gy. After adding a ferroptosis inhibitor, liproxstatin-1, to the irradiated mice, the amount of cell damage and cells going into ferroptosis was reduced. This was linked to decreased ROS levels, decreased pro-inflammatory cytokines, and increased GPX4 levels [165] (Figure 3).

Consistent with the above-discussed findings, different studies were initiated to develop new treatments to diminish RILI. One study focussed on NVP-AUY922, a novel heat shock protein 90 (HSP90) inhibitor as a radioprotector. Using lung epithelial cells and in vivo studies in combination with ferrostatin-1, the central role of ferroptosis in RILI was confirmed. Subsequently, treatment with NVP-AUY922 induced similar effects as ferrostatin-1. The effects were mainly attributed to elevated levels of GPX4 [166] (Figure 3).

Another study identified a novel axis activated after RT and involved in ferroptosis induction, namely the PIEZO1/Ca^2+^/calpain/VE-cadherin signaling pathway. Single high dose RT (15 Gy) induced upregulation of PIEZO1, a calcium influx modulator, on lung endothelial cells. Upregulated levels were linked with increased calcium levels, leading to increased activation of the calcium-dependent calpain, which in turn degraded VE-cadherin, leading to pulmonary endothelial cell ferroptosis. Hence, inhibitors of this pathway such as GsMTx4 (PIEZO1 inhibitor) or PD151746 (calpain inhibitor), prevented ferroptosis and reduced RILI [167] (Figure 3) (Table 2).
ijms-25-03641-t002_Table 2Table 2Normal tissue toxicity after radiation. Parameters investigated for normal tissue toxicity determination. The yellow, green, pink and purple color refer to the colors used in Figure 3.Tissue of InterestIrradiation Source/DoseParameters Investigated**Lung**TBI γ-Ray (17 Gy)Collagen depositionThorax irradiation X-ray (15 Gy) [164]Szapiel score, Ashcroft score and hydroxyproline contentThorax irradiation X-ray (15 Gy) [165]Lung injury score (Szapiel)Thorax irradiation X-ray (10 Gy) [166] Collagen deposition and % apoptotic cells**Gastro-intestinal system**TBI γ-ray (15 Gy) [160] Villi height, % apoptosis and proliferationTBI 60CO γ-source (9 Gy) [168]Villus height, crypt depth, proliferation, crypts per circumference, apoptosis, DNA damage, and oxidative stressTBI (9.25 Gy) [169]% Apoptosis and epithelial barrier disruption**Skin**Electron beam (45 Gy) [162]Number of skin appendages and epidermis depthUVB lamp [163]Epidermal and dermal thickness**Hematopoietic system**TBI γ-Ray (4 Gy) [160]Red blood cell count, white blood cell count, hemoglobin, platelets, percentage of lymphocytes, and percentage of neutrophil granulocytesWhole body irradiation 137Cs source (10 Gy) [169]Bone marrow hemorrhage, hepcidin content, iron content, white blood count, lymphocytes, red blood cell count, platelets, and hemoglobinTotal body irradiation 137Cs source (8 Gy–10 Gy) [170]Histology of the bone marrow, red blood cell count, white blood cell count, lymphocytes, monocytes, and platelets

### 5.4. Ferroptosis Has Been Connected to Gastrointestinal Epithelial Injury

The first evidence suggesting a link between radiation-induced intestinal injury (RIII) and ferroptosis was presented by Xie et al., who proposed a new potential radioprotector, namely (-)-epigallocatechin-3-gallate (EGCG). Human intestinal epithelial cells (HIEC) were treated with EGCG and irradiated (9 Gy). Cell viability clearly improved after treatment, which was associated with decreases in DNA damage, oxidative stress, apoptosis, and ferroptosis. The radioprotective role of EGCG was attributed to the activation of NRF2, which led to downstream triggering of SLC7A11 and GPX4. Confirmation was achieved in a mice model undergoing TBI (9 Gy). Prolonged survival and inhibition of body weight loss were observed in EGCG-treated mice, linked to retainment of the normal crypt-villus structure and active intestinal stem cells [168] (Figure 3).

Additionally, a connection between the gut microbiome and the influence on ferroptosis induction, and thereby normal tissue toxicity, was established. The presence of *P. aeruginosa*, an opportunistic bacterial pathogen, within the gut during TBI (9.25 Gy) decreased the survival of mice and aggravated intestinal injury. The pathogen-induced “theft-ferroptosis”, disrupts the gut epithelium and opens the door for an intestinal infection, bacteremia, and sepsis-induced death following TBI. A 15-lipoxygenase inhibitor, baicalein, was able to reverse the increased mortality induced by elevated levels of *P. aeruginosa* (Figure 3). Hence, inhibition of ferroptosis mitigated the injury induced by specific gut microbiota during TBI via elevation of GPX4 and inhibition of 15-lipoxygenase [169] (Table 2).

### 5.5. Ferroptosis Influences Hematopoietic Injury after Ionizing Radiation

The connection between hematopoietic acute radiation syndrome and ferroptosis was established by Zhang et al. Gamma-irradiation (4 Gy or 8 Gy) of mice induced bone marrow bleeding, leading to increased levels of iron within the bone-marrow microenvironment. These increases in iron content triggered the induction of ferroptosis of the granulocyte-macrophage hematopoietic progenitor cells, resulting in decreased levels of white blood cells. Since ferroptosis induction occurred within the progenitor cells, the potential of ferroptosis inhibitors to attenuate the radiation-induced injury was evaluated. Ferrostatin-1 and LDN193189 (iron inhibitor) significantly expanded the life span of gamma-irradiated mice [170] (Figure 3). Additionally, it was shown that gamma-irradiation increased the iron levels and decreased ACSL4 and lipoxygenase 15 levels, which enhanced lipid peroxidation and thereby induced ferroptosis. Ferrostatin-1 treatment was able to counteract these effects, mitigating the damage induced by radiation on the red and white blood cells [171] (Table 2).
Figure 3Influence of ferroptosis blockers on normal tissue toxicity. Different ferroptosis inhibitors have been linked to ameliorating normal tissue toxicity after RT. In this figure, the specific inhibitors are depicted in correlation to the organ of interest.
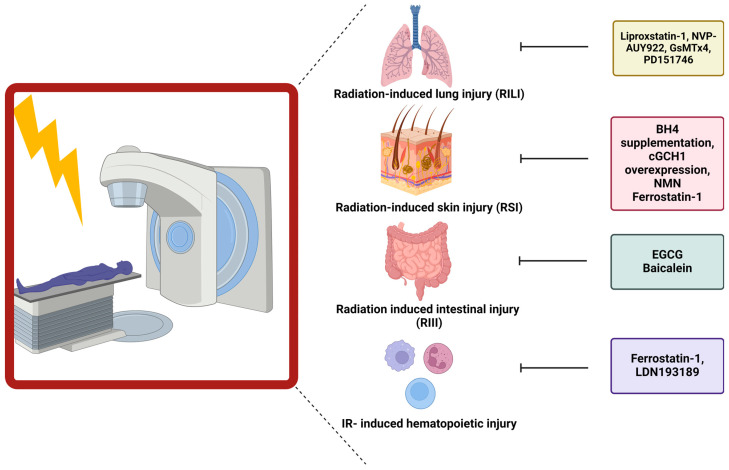


## 6. Association between FLASH-RT and Ferroptosis

FLASH-RT is a novel treatment modality, attracting significant attention across all fields of radiotherapeutic research. The use of ultra-high dose-rates to deliver radiation is shown to achieve an identical amount of tumor killing while sparing the normal tissue present within the radiation frame. However, the exact underlying mechanism responsible for the FLASH effect remains elusive. The first connection between FLASH-RT and ferroptosis was suggested by Vilaplana-Lopera et al. [172]. An inverse relationship between dose-rate and lipid peroxidation was established in the early 1990’s [173]. As a logical consequence, less ferroptosis would be present after FLASH-RT compared to conventional RT, mitigating the RT-induced injury. Additionally, the FLASH-effect has been attributed to the depletion of oxygen. Ultra-high dose-rates rapidly deplete oxygen, leading to a more pronounced effect in normal tissue since most solid tumors are already hypoxic. This depletion of oxygen can inhibit the lipid chain reaction, the main process driving ferroptosis, and consequently leading to a decrease in the levels of lipid ROS. Therefore, FLASH possibly inhibits ferroptosis within normal tissue [172].

## 7. Conclusions and Perspectives

The following question is of utmost importance: is ferroptosis a friend or an enemy during RT? A clear link between ferroptosis and RT has been established in multiple solid tumors in vitro, in vivo, and in patient samples. Since the discovery that IR induces ferroptosis, interest in using FINs as radiosensitizers has been growing. Research investigating ferroptosis has recently increased, as it offers a promising avenue for anti-cancer treatment. Given that the limited number of ferroptosis studies includes hypoxic conditions, it is warranted to first discover the exact working mechanism of FINs under hypoxic conditions before considering their clinical translation as radiosensitizers.

During the course of this review, it became clear that sensitivity toward FINs is context-dependent. Consequently, the development of specific biomarkers distinguishing between responsive cells and resistant cells is of interest. A range of cellular molecules have already been described as biomarkers for ferroptosis sensitivity, f.e. NADPH, p53, GPX4, and ACSL4. However, the applicability of specific biomarkers in specific cancer types needs further elucidation.

Most of the acknowledged FINs are lab-designed molecules with low bioavailability in an in vivo setting. A possible solution is the use of FDA-approved drugs, which can be repurposed as radiosensitizers in the clinic. Several FDA-approved drugs have already been linked with the induction of ferroptosis, although knowledge of the exact contribution of FINs in RT-response is still limited. More elaborate investigation is recommended before this strategy can be used to overcome the problem of poor pharmacological profiling.

Clinical translation of FINs warrants more extensive research into the role of ferroptosis during normal tissue toxicity since recent reports indicate that ferroptosis plays a central role in the development of this major side effect. Investigation into the treatment timing and specific treatment delivery can provide possible solutions to this drawback. To obtain significant tumor control, treatment with FINs is recommended to be administered before and during IR. Supplementation of ferroptosis inhibitors can be initiated after the radiation regimen is completed to diminish or prevent normal tissue toxicity. Additionally, targeted delivery of FINs to the tumor cells could be of interest to reduce adverse effects. Albeit the use of FINs as radiosensitizers is promising, further research into the effects under hypoxic conditions and their consequences on normal tissue is essential to make clinical translation feasible.

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
