# Peer review of "Ferroptosis: Frenemy of Radiotherapy"

_ijms, 2024, doi:10.3390/ijms25073641_

Round 1

Reviewer 1 Report (New Reviewer)

Comments and Suggestions for Authors

The submitted review article introduced a useful aspect but requires the following points:

The written text should be simpler and more fluent

The main drawback of this study is the lack of forms to better describe the process

Comments on the Quality of English Language

This text needs editing and should be written more simply

Author Response

Reviewer 2 Report (New Reviewer)

Comments and Suggestions for Authors

Ferroptosis is explained well in the review, but adding histological images at the light and electron microscope level will make the subject more understandable. It was observed that microscope images of cell death patterns were not included in the explanation. I would like to re-evaluate the article after the images are added.

Author Response

Please see attachement. 

Reviewer 3 Report (New Reviewer)

Comments and Suggestions for Authors

Dear Author, 

manuscript present in interesting way the theme of ferroptosis and present also clinical aspects of this mechanisms. Some new aspects of ferroptosis in onclogy are discuss. The references contain as many as 198 items, this is worth for nothing. I recommend correct figure 1: background it is too dark and there is a lack of molecular effects in the sheme. Also the font of citing references in the manuscript text seems to need modification. 

Author Response

Reviewer 4 Report (New Reviewer)

Comments and Suggestions for Authors

It is an interesting topic. The authors comprehensively describe ferroptosis in the context of radiotherapy. The article is clear and well structured. Furthermore, I make minor suggestions to improve the clarity and quality of the article.

1. It is advisable to consider adding more images that would more clearly explain the mechanisms described in the text at the molecular level.

2. The images are of insufficient quality, with only a small magnification they are not readable enough. For example in Fig. 1 takes up too much free space.

Round 2

Reviewer 1 Report (New Reviewer)

Comments and Suggestions for Authors

This paper is accepted

Reviewer 2 Report (New Reviewer)

Comments and Suggestions for Authors

It is appropriate for the study to be published as a result of the corrections made by the author.

This manuscript is a resubmission of an earlier submission. The following is a list of the peer review reports and author responses from that submission.

Round 1

Reviewer 1 Report

Comments and Suggestions for Authors

[1]     Lines 234-235 - Briefly explain the reasons for intracellular ROS aggregation caused by hypoxia.

[2]     Lines 320 - 4.4-4.10these drugs with iron death induction and radiosensitizing properties would be more appropriate as the subheading of 4.3

[3]     Lines 342-344 - The examples given in 342-344 do not have a strong correlation with ferroptosis. A more relevant example could be used to introduce the DR strategy.

[4]     Lines 381-382 - The expression of the sentence is not particularly clear. Use a more appropriate word instead of 'limited'.

[5]   Lines 602-637 - 5.2 "Role of ferroptosis in radiation induced lung injury", 5.3 "Link between ferroptosis and radiation induced skin injury" Subtitles need to be repositioned, starting with 5.1 "Interplay between ferroptosis and total body injury" throughout the body, followed by 5.2"Link between ferroptosis and radiation induced skin injury" skin. Then 5.3"Role of ferroptosis in radiation induced lung injury" will make more logical.

[6]      Lines 694 - 5.6 This subtitle and content are not so strongly related to the main title of Part 5, and it may be considered to be discussed separately.

[7]      Cited the “Review of the Role of Ferroptosis in Testicular Function” in the Introduction to ferroptosis.

Author Response

Point-by- point reply Reviewer 1

We would like to thank the reviewer to take the time to thoroughly revise our manuscript and the acknowledgment of its value.

[1]  Lines 234-235 - Briefly explain the reasons for intracellular ROS aggregation caused by hypoxia.

Thank you for this valuable suggestion. The addition of the reasons why intracellular ROS is increased under hypoxic conditions is of added value to the manuscript. Therefore, the following text was added in line 245-250: “It is well-known that hypoxia results in an increase in intracellular ROS, due to the prolongation of the lifetime of the ubisemiquinone radical, a molecule generated at complex III of the mitochondria, which can interact with oxygen, leading to the generation of ROS. Additionally, inducible nitric oxide synthase (iNOS) is a hypoxia response gene, hence gets upregulated under hypoxic conditions, leading to an increased generation of nitric oxide (NO).”

[2]  Lines 320 - 4.4-4.10: these drugs with iron death induction and radiosensitizing properties would be more appropriate as the subheading of 4.3

The subheading of 4.3 is adjusted in the manuscript to “Drugs with ferroptosis inducing capacities and radiosensitizing properties.” On line 417 of the manuscript.

[3]  Lines 342-344 - The examples given in 342-344 do not have a strong correlation with ferroptosis. A more relevant example could be used to introduce the DR strategy.

Thank you for this suggestion, the reviewer is correct. We removed the example of sildenafil (lines 342-344) and focused on the example of thalidomide, which is repurposed in the field of oncology (line 352-355). Furthermore, two examples of drug repurposing in the field of ferroptosis were added, namely sulfasalazine and sorafenib, by addition of the following text: “Moreover, the use of FDA approved drugs as FINs is under intensive investigation. Two pharmacological agents, sulfasalazine and sorafenib were initially developed for the treatment of rheumatoid arthritis and hepatocellular carcinoma, respectively. Nevertheless, it has been elucidated that these drugs contain system xC- inhibitory capacities, inducing ferroptosis within a range of cancer cells. The link with RT will be described in-depth in the following paragraphs.” (line 355-360).

[4]  Lines 381-382 - The expression of the sentence is not particularly clear. Use a more appropriate word instead of 'limited'.

We apologize that the sentence was not completely clear. The word limited was replaced by restricted and the sentence was rewritten to: “Of note, the evidence linking the radiomodulatory effect of DSF to ferroptosis induction is currently restricted to a single publication” in line 392-394.

[5]   Lines 602-637 - 5.2 "Role of ferroptosis in radiation induced lung injury", 5.3 "Link between ferroptosis and radiation induced skin injury" Subtitles need to be repositioned, starting with 5.1 "Interplay between ferroptosis and total body injury" throughout the body, followed by 5.2"Link between ferroptosis and radiation induced skin injury" skin. Then 5.3"Role of ferroptosis in radiation induced lung injury" will make more logical.

We agree with the reviewer that starting from a more general perspective and afterwards go to the individual organs would create a more natural flow. The different subtitles were repositioned within the text. “Link between ferroptosis and radiation induced skin injury” became 5.2 in the manuscript (line 614), while “Role of ferroptosis in radiation induced lung injury” Became 5.3 in the manuscript (line 638).

[6]   Lines 694 - 5.6 This subtitle and content are not so strongly related to the main title of Part 5, and it may be considered to be discussed separately.

The reviewer is correct in stating that FLASH-RT is not strongly related to normal tissue toxicity after RT in correlation to ferroptosis. The FLASH-effect attracted a lot of attention over the past years, since it reduces normal tissue toxicity while sparing the tumor control effect by using ultra high dose rates. Therefore, this section was included under the subtitle adverse events. It is indeed more fitting to include a separate paragraph discussing FLASH-RT. Hence, “Association between FLASH-RT and ferroptosis” was replaced to heading 6 in the manuscript (line 709).

[7]   Cited the “Review of the Role of Ferroptosis in Testicular Function” in the Introduction to ferroptosis.

Thank you for this valuable suggestion. Besides oncology, ferroptosis plays an important role in other research fields, such as neurological diseases [21], organ injuries [22], the reproductive system [23], etc. However this is beyond the scope of this review. These additional sentences and references were added to the introduction of ferroptosis (line 77-80).

Reviewer 2 Report

Comments and Suggestions for Authors

In the present manuscript author try to describe novel concept of radiotherapy with the ferroptosis and control the tumour progression. Over all manuscript looks good, but few concerns are need to clear before further proceeding. Such as, ferroptosis clinical relevance needs to understand with radiotherapy. Furthermore, this type of combi regimen how help for mutation driven cancer need to describe. Any research update for novel molecule which are induces ferroptosis in cancer cells. How ferroptosis overcome the drug resistance in cancer therapy not mentioned properly. Also, author need to describe how these type therapy beneficial over modern therapy.       

Author Response

 Point-by- point reply Reviewer 2

In the present manuscript author try to describe novel concept of radiotherapy with the ferroptosis and control the tumour progression. Over all manuscript looks good, but few concerns are need to clear before further proceeding.

We would like to thank the reviewer to take the time to thoroughly revise our manuscript and the acknowledgment of its value. We are very pleased to read that overall, the manuscript looks good to this reviewer’s opinion.

  • Ferroptosis clinical relevance needs to understand with radiotherapy.

We understand the concern of the reviewer that the relevance of ferroptosis in the clinic after radiotherapy is not completely comprehensive. Since ferroptosis is a relatively new type of regulated cell death, firstly described in 2012, the exact contribution of ferroptosis after radiotherapy is not completely elucidated. However, since 2019 the first preclinical publications linking ferroptosis to radiotherapy have been published by Lang et al. [1], Ye et al. [2] and Lei et al. [3]. Out of these publications, it became clear that lipid peroxidation and ferroptosis induction is present after irradiation of different types of cancer cells (lung cancer, fibrosarcoma, melanoma, breast cancer, and esophageal carcinoma). Furthermore, Lei et al. performed immunohistochemical staining’s on esophageal tumor tissue of patients before and after radiotherapy and could identify that an increase in 4-HNE (an end product of lipid peroxidation) was present after radiotherapy. The intensity of 4-HNE staining was linked to disease-free survival after radiotherapy. Hence, a clear correlation between ferroptosis induction after radiotherapy and a better disease-free survival was identified. In conclusion, Lei et al. firstly demonstrated a clinical link between ferroptosis and radiotherapy in patient samples and the importance on the disease progression.

To illustrate the clinical relevance in the manuscript, we added the following sentences: “Moreover, the first clinical link between ferroptosis and radiotherapy was elucidated by Lei et al. The degree of ferroptosis induction in samples of esophageal cancer patients showed a correlation with disease-free survival in response to RT.” (Line 331- 334 of the manuscript.)

  • Furthermore, this type of combi regimen how help for mutation driven cancer need to describe.

Thank you for the suggestion to clarify the relationship between ferroptosis and the mutational status of cancer cells. Unfortunately, exact correlations between ferroptosis sensitivity and the mutational status of cancer is still under thorough investigation, with conflicting results being published. Nonetheless, we attempted to summarize the existing knowledge below and adapted the review further.

The firstly discovered ferroptosis inducer, erastin, was predicted to induce ferroptosis in cancer cells carrying a RAS-mutation (line 69-70). Despite this, a lot of controversy exist about the susceptibility of cancer cells to ferroptosis and the correlation with the mutational status of the cells. For example, some reports nowadays have stated that carrying a RAS-mutation is not significant enough to sensitize cancer cells to ferroptosis [4]. Hence, other genes and mutations may play a role in this process. The following sentence was added to the manuscript to make this concept more comprehensive: “However, still a lot of controversy exists about the mutational status of cells in relation to ferroptosis sensitivity.” (Line 72-73).

Moreover, p53 activity has also been linked to the sensitivity of cancer cells towards ferroptosis. Upregulation of p53 is directly correlated to suppression of SLC7A11 (line 234-236), making cancer cells more susceptible to ferroptosis. However, in colorectal cancer p53 led to the translocation of DDP4 to the nucleus, making it lose its ability to induce lipid ROS (line 236-237).

Hence future research investigating the exact role of different driving- cancer mutations on ferroptosis sensitivity is warranted before any stringent scientific conclusions can be drawn.

  • Any research update for novel molecule which are induces ferroptosis in cancer cells.

Indeed, this is a very interesting avenue to explore. One example of novel molecules can be nanomaterials/nanoparticles, which have attracted a lot of interest over the past decade (line 340-343) and are identified as the most novel ferroptosis inducers [5]. Iron-based nanoparticles are particularly interesting since they can directly trigger the Fenton reaction, however this is beyond the scope of our review.

  • How ferroptosis overcome the drug resistance in cancer therapy not mentioned properly.

We apologize that this was not clearly described in the review.

The opportunity to use ferroptosis to overcome drug resistance is a topic of great interest, however in this review we are focusing on radiotherapy and radioresistance.

For further clarification: Drug resistance can develop due to tumor heterogeneity, the presence of cancer stem cells, the tumor microenvironment, and the interaction between drugs and cancer [6]. The development of drug resistance to classical chemotherapy (like cisplatin), targeted therapies and immunotherapy have been extensively described. Nevertheless, preclinical studies have found the first evidence that ferroptosis inducers are capable to overcome resistance to cisplatin in ovarian cancer and gastric cancer. Additionally, ferroptosis is capable of inhibiting epithelial to mesenchymal transition (EMT) and thereby can interfere with the development of drug resistance against targeted therapies. Lastly, addition of ferroptosis inducers can synergize with immunotherapy. Since, cytotoxic T- cells release interferon gamma, leading to an activation of STAT1 and inhibition of xCT, enlarging the proportion of cells going into ferroptosis [6]. We mentioned that ferroptosis research and its influence on drug resistance is thriving (line 77), however an in-depth explanation is beyond the scope of our review.

  • Also, author need to describe how these type therapy beneficial over modern therapy.

Thank you for this remark. The answer to the question on how this type of therapy could be beneficial over the existing treatments is intertwined with the first question on the clinical relevance of ferroptosis in relation to radiotherapy. The first evidence of the importance of ferroptosis induction in radiotherapy responses has been made clear, but the clinical relevance and benefits of this type of therapy in patients is still under investigation. The combination of ferroptosis induction and radiotherapy could be beneficial over traditional radiation therapy, since it can work in an additive or synergistic manner (line 329-331). Reverting radioresistant tumors into more vulnerable tumors upon radiotherapy, while maintaining/diminishing toxicity for patients.

However, as mentioned in the manuscript (section 5) caution should be taken before clinical translation is possible, since ferroptosis might play a role during normal tissue toxicity development after radiotherapy.

We want to encourage researchers/clinicians to ponder about the benefits and drawbacks of ferroptosis-related therapy. In order to be able to make a proper risk assessment before bringing this type of therapy to the clinic, as annotated in the conclusion section.

  1. Lang, X.; Green, M.D.; Wang, W.; Yu, J.; Choi, J.E.; Jiang, L.; Liao, P.; Zhou, J.; Zhang, Q.; Dow, A.; et al. Radiotherapy and Immunotherapy Promote Tumoral Lipid Oxidation and Ferroptosis via Synergistic Repression of SLC7A11. Cancer Discov 2019, 9, 1673–1685, doi:10.1158/2159-8290.CD-19-0338.
  2. Ye, L.F.; Chaudhary, K.R.; Zandkarimi, F.; Harken, A.D.; Kinslow, C.J.; Upadhyayula, P.S.; Dovas, A.; Higgins, D.M.; Tan, H.; Zhang, Y.; et al. Radiation-Induced Lipid Peroxidation Triggers Ferroptosis and Synergizes with Ferroptosis Inducers. ACS Chem. Biol. 2020, 15, 469–484, doi:10.1021/acschembio.9b00939.
  3. Lei, G.; Zhang, Y.; Koppula, P.; Liu, X.; Zhang, J.; Lin, S.H.; Ajani, J.A.; Xiao, Q.; Liao, Z.; Wang, H.; et al. The Role of Ferroptosis in Ionizing Radiation-Induced Cell Death and Tumor Suppression. Cell Res 2020, 30, 146–162, doi:10.1038/s41422-019-0263-3.
  4. Andreani, C.; Bartolacci, C.; Scaglioni, P.P. Ferroptosis: A Specific Vulnerability of RAS-Driven Cancers? Front. Oncol. 2022, 12, 923915, doi:10.3389/fonc.2022.923915.
  5. Yu, H.; Yan, J.; Li, Z.; Yang, L.; Ju, F.; Sun, Y. Recent Trends in Emerging Strategies for Ferroptosis-Based Cancer Therapy. Nanoscale Adv. 2023, 5, 1271–1290, doi:10.1039/D2NA00719C.
  6. Nie, Z.; Chen, M.; Gao, Y.; Huang, D.; Cao, H.; Peng, Y.; Guo, N.; Wang, F.; Zhang, S. Ferroptosis and Tumor Drug Resistance: Current Status and Major Challenges. Front. Pharmacol. 2022, 13, 879317, doi:10.3389/fphar.2022.879317.

Reviewer 3 Report

Comments and Suggestions for Authors

This appears to be a thorough and well-written submission. I am not an expert in radiotherapy or oncology more generally: however, there are some general observations I have about this manuscript proposal. 

The review is written in an unstructured format and shows no evidence of the establishment of a protocol prior to undertaking the review procedures. There is no Methods section or description of how the review was conducted. It is essential that all review types establish a protocol, articulate their protocol, and present their findings in a transparent and repeatable manner. Reproducibility and transparency are the bedrock of science, without which the value and benefit are lost. This reviewer would suggest the authors define a protocol following PRISMA or one of its modifications and re-execute their review.   

Author Response

Point-by- point reply Reviewer 3

This appears to be a thorough and well-written submission. I am not an expert in radiotherapy or oncology more generally: however, there are some general observations I have about this manuscript proposal. 

We would like to thank the reviewer to take the time to thoroughly revise our manuscript and the acknowledgment of its value. We are very pleased to read that the manuscript is thorough and well-written according to this reviewer’s opinion.

The review is written in an unstructured format and shows no evidence of the establishment of a protocol prior to undertaking the review procedures. There is no Methods section or description of how the review was conducted. It is essential that all review types establish a protocol, articulate their protocol, and present their findings in a transparent and repeatable manner. Reproducibility and transparency are the bedrock of science, without which the value and benefit are lost. This reviewer would suggest the authors define a protocol following PRISMA or one of its modifications and re-execute their review.   

The reviewer is absolutely correct in stating that reproducibility and transparency are the bedrock of science. In the case of a systematic review/ meta-analysis where the main goal is to find an answer on a specific clearly-formulated research question, the use of a systematic method to minimize bias is essential [1].

In our review, we aimed to encourage researchers/clinicians to ponder about the benefits and drawbacks of ferroptosis-related therapy in order to be able to make a proper risk assessment before bringing this type of therapy to the clinic. We did not focus on a clear question about a specific patient group or the outcomes of interest. Therefore, we opted to write a literature/narrative review to collect a comprehensive summary of research data on the connection between ferroptosis and radiotherapy, ferroptosis and drug repurposing, and ferroptosis and normal tissue toxicity.

In the medical field, the narrative review is a common format. This type of review is considered an important educational tool, following a less strict format compared to a systematic review. Reporting search terms, used databases, inclusion and exclusion criteria and a strict methodology is not required [2].

Since we do not have a defined research question following the PICO format (population, intervention, control and outcome) [1] necessary for the PRISMA guidelines it is impossible to formulate eligibility criteria, (inclusion/exclusion criteria) [3].

In order to meet the concerns of the reviewer, we performed an extensive search of literature reviews that have included a PRISMA protocol, but did not find any applicable examples. Furthermore, the guidelines of this journal do not specify the need for a method section.

The protocol that we established before starting to write this review was as followed:

  • Identified a gap in the current literature where both the advantages and drawbacks of ferroptosis induction after radiotherapy treatment are described.
  • The following Keywords/ MESH terms were used to search for similar reviews to determine the value of our idea: Ferroptosis, Radiotherapy, Hypoxia, Normal tissue toxicity.
  • A lay-out of the review was drafted:
  1. Short overview of the separate components that will be discussed throughout the review.
  2. Discussion of combinational effects of the separate components.
  3. (Possible) clinically relevant interventions to increase the effectiveness of radiotherapy in solid tumors.
  4. (Possible) adverse events of this type of therapy in the clinic with regard to normal tissue toxicity.
  5. Newest revelation in reducing normal tissue toxicity in radiation research.
  6. General conclusion to encourage researchers/clinicians to ponder about the benefits and drawbacks of ferroptosis-related therapy.

The following Keywords/ MESH terms were used to search for relevant publications: ferroptosis, radiotherapy, drug repurposing, hypoxia, normal tissue toxicity, total body irradiation, radiation-induced lung injury, radiation-induced skin injury, radiation-induced gastrointestinal injury, radiation-induced hematopoietic injury and FLASH-RT.

Two different databases were searched: PubMed and Web Of Science. Additionally, other articles cited in the previous ones extracted and of interest in this topic were included within the review, until reaching a saturation point.

Despite all this, we acknowledge that a narrative review has some limitations compared to a systematic review, such as biases due to the lack of a PRISMA method section and the influence of the author’s perspective on the outcomes [2].

  1. Charrois, T.L. Systematic Reviews: What Do You Need to Know to Get Started? CJHP 2015, 68, doi:10.4212/cjhp.v68i2.1440.
  2. Jahan, N.; Naveed, S.; Zeshan, M.; Tahir, M.A. How to Conduct a Systematic Review: A Narrative Literature Review. Cureus 2016, doi:10.7759/cureus.864.
  3. Page, M.J.; McKenzie, J.E.; Bossuyt, P.M.; Boutron, I.; Hoffmann, T.C.; Mulrow, C.D.; Shamseer, L.; Tetzlaff, J.M.; Akl, E.A.; Brennan, S.E.; et al. The PRISMA 2020 Statement: An Updated Guideline for Reporting Systematic Reviews. BMJ 2021, n71, doi:10.1136/bmj.n71.

Round 2

Reviewer 1 Report

Comments and Suggestions for Authors

accept

Author Response

We would like to express our sincere gratitude for taking the time to thoroughly evaluate our work. The proposed adaptations have significantly enhanced the overall quality of this review. We would like to thank you for acknowledging the value of our review in its current state and for recommending its acceptance.

Reviewer 2 Report

Comments and Suggestions for Authors

Now manuscript will be accepted for publication 

Author Response

(The authors gave the same response as above.)

Reviewer 3 Report

Comments and Suggestions for Authors

This reviewer appreciates the authors' response. The authors' explanation has helped this reviewer understand the more broad intent of the authors' review. Nonetheless, this review would most appropriately be written and considered as a scoping review, which is less restrictive than a systematic review but more structured and of a more repeatable design than a traditional narrative review. This reviewer suggests that the authors revise their manuscript proposal accordingly. The guidelines for a scoping review are available here: http://www.prisma-statement.org/Extensions/ScopingReviews?AspxAutoDetectCookieSupport=1

Author Response

First, I would like to express my gratitude for taking the time to thoroughly evaluate our work. Constructive criticism is essential for academic growth, and we appreciate the feedback provided during the first revision round. 

However, after careful consideration, we find ourselves in disagreement with one remaining remark made by the reviewer. While we respect the reviewer's perspective and understand the importance of rigorous peer review, we consider the suggestion of completely rewriting this review in the format of a scoping review excessive. 

The reviewer criticizes the review for lacking guidelines. However, it is important to note that the submission guidelines for the special issue 'Regulation and Targeting of Ferroptosis in Tumor and Beyond' in IJMS do not explicitly mandate a method section for narrative reviews. We believe our manuscript adheres to the provided guidelines, as acknowledged by both reviewer one and reviewer two. Furthermore, within this special issue, narrative reviews with a similar layout have already been published. 

We kindly request that the reviewer reconsider this point. The reviewer has acknowledged that this review was thorough and well-written and provided a beneficial overall score. We firmly believe that our review makes a meaningful contribution to the literature in our field. 

Thank you for your time and consideration. 

Round 3

Reviewer 3 Report

Comments and Suggestions for Authors

This reviewer appreciates the authors' exasperation at the suggestion to restructure their manuscript proposal as a scoping review, but this reviewer has personally received this same type of feedback from peer review, which was enforced by the editor-in-chief in my case. An esteemed former mentor of this reviewer's, who was a very senior academician, had the same experience. This reviewer's suggestion was not made without careful thought and remains unchanged. It will make the manuscript proposal better, and the reviewer encourages the authors to carefully implement it.